# Tourism Dynamics and Sustainability: A Comparative Analysis between Mediterranean Islands—Evidence for Post-COVID-19 Strategies

Giovanni Ruggieri [1,*] and Patrizia Calò [2]

1   Department of Economics, Business and Statistics (SEAS), University of Palermo, 90128 Palermo, Italy
2   Observatory on Tourism in the European Islands—OTIE, 90139 Palermo, Italy; research@otie.org
*   Correspondence: giovanni.ruggieri@unipa.it

**Abstract:** Tourism may not sustainably support territories with limited natural resource stock such as islands. The volume of visitor arrivals and the industry investments can increase the pressure even beyond sustainable levels. There is an evident and unresolved tension between these two great polarities, sustainability and economic growth driven by tourism. The aim for policymakers is to find an acceptable equilibrium between these two dimensions. This paper investigates tourism evolution between 2007 and 2019 in 15 Mediterranean islands, comparing tourism pressures through statistical indicators. The analysis will compare tourism demand and supply trends in these contexts. The performances will be evaluated to identify the islands' positioning between sustainability needs and tourism development opportunities while considering post-COVID-19 challenges.

**Keywords:** tourism; islands; impact; economic development; sustainability

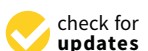



## 1. Introduction

The consideration of tourism as a development driver is still under discussion because the efforts to enhance local benefits and competitiveness in tourism seem controversial from a sustainable perspective. Despite this consideration, the potential economic growth of tourism is documented in the international literature, as highlighted in several recent studies. In general, when tourist activity grows, visitors increase and spend more money in a destination, leading to an increase in the GDP and economic growth [1–11].

With regard to insular contexts, the need to consider the peculiarity of these territories emerges. Tourism in islands is not a solved question because islands have a limited natural resource stock, so the increase in visitor arrivals can put pressure on the use of these to their viability limit, even beyond sustainable levels. Studies on the impact of tourism on island destinations worldwide have shown both positive and negative externalities generated by tourism in these contexts [12,13]. The increase in tourism flows could have unexpected detrimental impacts on environments and local communities, derived from the excess of tourism, called overtourism [14–16]. Monitoring tourism impacts is fundamental to avoiding negative effects on the environment and residents [17] and finding new opportunities for the expansion of local industries [18–21]. In terms of sustainability [22] (1–20), for coastal or island areas it is not easy as they are territorial targets for significant tourism flows. In this way, Mediterranean islands' environmental and cultural images can act as magnets for attracting many tourists (i.e., overnight visitors). However, the arrival of consistent tourist flows could alter the fragile insular ecological equilibrium, negatively affecting those natural and cultural resources that have initially aroused tourists' interest in the knowledge of that place and ultimately could cause the displacement of tourists to the islands. That is the "paradox of tourism in the islands" [23] (131–143), and this is even more significant in the Mediterranean space. Tourism appears as an essential part of the local economy [13], being perceived generally as one of the few economic development

opportunities available in the insular context and the only natural economic alternative (at the production level, economic activity, and income) capable of responding to the socio-economic needs of its inhabitants. Given that, the need to contain and eliminate negative effects on the environment and residents emerges.

The unresolved tension between these two great polarities, sustainability and economic growth, is still far from an acceptable equilibrium. In this sense, a current line of critical thinking [24,25] rejects the use of the term "sustainable tourism", suggesting that its use can be instrumented by political actors whose fundamental objective is somehow "green" but is mainly economic growth. Adequate implementation of sustainable tourism [26] must emphasize the systematic management of environmental degradation, the generation of economic benefits for the receiving communities, and residents' perception [27–29].

This paper will explain how sustainable tourism was an accepted and practical reality in Mediterranean islands and what tourism development could be undertaken in these contexts. A sample of the Mediterranean island territories belonging to the European Union was included in the proposed analysis to explore this possibility.

After examining the features of tourism demand and supply in each insular context, the paper analyzes four indicators comparing the results in the different observed islands. The positioning of each context, depending on the combination of the values obtained by the indicators selected in the two years, is observed. This allows us to also consider the evolution over the period observed.

The proposed analysis concerns the performances recorded before the COVID-19 pandemic. The effects of COVID-19 on the economy of the Mediterranean islands, especially in terms of tourism, could be defined as disastrous, with tourist activity on many islands having been reduced by almost 80%. The combined effect of restrictions (curfews, lockdowns, closing of theaters and discos, the closure of hotels and restaurants) with travel difficulties (border closures, shortage of air and maritime connections, airport closures), and the fear of being infected or becoming infected has caused a tremendous crash of the demand, causing a severe economic crisis worldwide and especially in islands, whose economy depends mainly on tourism. From a sustainability perspective, COVID-19 should be an opportunity to rethink tourism in the Mediterranean islands more consciously, by achieving a balanced equilibrium between the policies to increase the tourism industry and public policy to contain and protect the islands' territories.

## 2. Insularity Condition and Tourism

Isolation determines the islands' social, cultural, political, and economic life. Historically, being isolated from the outside world, the islands appeared to be considered autarkic societies, without social and economic dynamism and with few commercial relations. Hence this nineteenth-century idea of the islands as ultraconservative, immovable, and atavistic societies reluctant to change, whose distrustful island population hardly interacts with outsiders. This is a typical romantic idea, but its influence still continues today [28].

Separation and unavoidable "territorial discontinuity" affect the life of the islands by questioning their external accessibility, both for those who intend to leave and those who intend to enter the island, since the external mobilization of people can only be carried out through the air and maritime transport units. Likewise, uncertainty is generated in essential aspects of island life, such as providing necessities.

Insularity [30,31] requires a port infrastructure adequate to current needs and improved to meet demand expansions. Ports are needed for the reception of vessels and must be equipped with means for loading, disembarking, and storing goods, with devices for customs control. Passengers' entry and exit must also be foreseen. Likewise, airports and other connected infrastructures are essential for the accessibility to islands and, from the tourist perspective, currently even more important. Itineraries established in island transport may be affected by adverse weather and maritime conditions for navigation or by specific over-demands and thus generate discomfort in the mobilization of ordinary users

and consumers. Moreover, critical marine phenomena destroy port facilities, coastal roads, and homes.

Therefore, insularity can be considered according to two complementary dimensions. The former is related to the physical vulnerability of the islands in spatial terms (isolation, small size or smallness, scarcity of resources) in relation to specific characteristics associated with the physical and geographical features of these contexts. This dimension is persistent in economic–commercial or economic development analysis on the islands. While the latter dimension, the "islandness" [31,32] presents a rather metaphysical cut, it reflects feelings common to all islanders based on the isolation inherent to the insular nature of islands, usually in line with solid senses of roots and community.

According to the former dimension, territorial discontinuity increases the costs of external supply products and export goods caused by the mobilization and storage of shipments and landings. In this respect, we talk about the costs of insularity, spawning over time a whole literature on the nature of such costs [12,33–35], on the way to measure, calculate, and evaluate them [36,37], and more recently, how to compensate the excess costs caused by remoteness, insularity, and ultra-peripherality of the island territories [38,39].

We wonder whether it might be more expensive to consume and produce on an island than to do it on the mainland. According to Manera and Garau [28], the natural environment where human activity takes effects and conditions it. For this reason, the costs of insularity are evident since the smaller the territory, the greater the cost of human activity. Moreover, the further the part is from world economic flows, the more the costs increase [28]. From this perspective, the cost of accessing the market is much higher in the case of island economies: if we consider the transport of goods, for example, this is between two and four times more expensive than on the mainland. For this reason, the transfer of raw materials, the higher costs of storing stocks, the degradation of perishable products, and delays due to adverse weather conditions are critical factors that directly affect the competitiveness of island productions [28].

All these factors related to insularity and the verification of their simultaneous presence in these territories have led to the emergence of insular vulnerability [40–42]. In their economic development process [1], the islands start from a situation characterized by a multiplicity of handicaps and physical, financial, and sociocultural weaknesses that cannot be avoided; therefore, a specific policy design is needed. The open debate in the European Union on insularity, its costs, and the way to face them are far from reaching a conclusion.

For island contexts, tourism represents, in this sense, the only policy option to overcome the structural constraints imposed by the small size of their economies and the insular physical conditions.

From an economic point of view, many islands have simply insufficient domestic market demand for a good or service to enable local firms to achieve any efficiencies or economies of scale. However, in the case of tourism, the demand is imported (incoming tourism), and thus, the market size can change and increase due to the possibility to attract external visitors. In this way, a local firm operating on an island could have a larger market than the local context for its goods and services. Then, they may begin to achieve economies of scale and efficiencies thanks to the tourist flows [43] (453–465). Therefore, island firms can face the problem of the small size of the local market thanks to the demand deriving from the incoming tourists. Moreover, tourists are high spending people, so the incomes for local enterprises will increase more than proportionally. Given that, insular economies are almost totally based on tourism and related activities.

Another condition that affects islands is the geographical distance, which limits the accessibility to a destination with consequences for tourism flows, which are consequently affected by the higher cost of transport and the difficulty to reach them. Then, also for tourism, the need to consider the costs of insularity in the economic development dynamics arises.

Island destinations represent a unique cluster, where tourism development and sustainability issues are connected and represent crucial aspects of the local economy and well-being [44].

### 3. The Survey

Islands are defined as natural land extensions surrounded by water above the water level at high tide [45] (147–154). This geographic element differentiates them and identifies them from other territorial realities (such as peninsulas, capes, or promontories). Then, both characteristics, isolation and separation, define the island's nature and the basis of its insular condition, i.e., the fact of being an island or "insularity", a defining characteristic of islands, based on isolation and geographic discontinuity. In the selection of the sample observed, we considered the definition of "island" provided by Eurostat [46] as follows:

- Have an area greater than 1 km$^2$
- Are at least 1 km away from the mainland.
- Do not have bridge connections to the mainland.
- Have a stable population of at least 50 people.

However, here the two islands of Cyprus and Malta, excluded by the European body since their respective capital cities fall within their territories, are analyzed.

This indicates a first difference between the institutional contexts examined, namely the Mediterranean: island states, autonomous regional islands, and coastal islands, which belong to a region situated on the mainland. Mediterranean islands were classified according to their size and density. The geographical dimension and population are not only featured from a geographical point of view, but they are issues from which tourism impacts cannot be separated.

In this survey a clusterization of islands and archipelagos according to the following four categories was carried out:

1.  Micro islands = 0 km$^2$ > island area < 1000 km$^2$.
2.  Small islands = 1,001 km$^2$> island area < 5,000 km$^2$.
3.  Medium islands = 5,001 km$^2$ > island area < 10,000 km$^2$.
4.  Large islands = island area > 10,001 km$^2$.

The sample of Mediterranean islands (Table 1) was observed in order to analyze the leading type of tourism and sustainability dimensions.

**Table 1.** Demography and territorial data.

| Islands/Arcipelago | Island Group | State | Area Km$^2$ | Population Density 2019 |
|---|---|---|---|---|
| Sicily islands | Large | Italy | 25,703 | 191 |
| Sardinia islands | | Italy | 24,090 | 67 |
| Cyprus | Medium | Cyprus | 9251 | 130 |
| Corse | | France | 8680 | 39 |
| Crete | | Greece | 8261 | 77 |
| Balearic Islands | Small | Spain | 4968 | 239 |
| Northeastern Aegean Islands | | Greece | 4260 | 54 |
| Evia | | Greece | 3662 | 52 |
| Ionian Islands | | Greece | 2443 | 83 |
| Dodekanisa | | Greece | 2393 | 80 |
| Cyclades | | Greece | 2267 | 53 |
| Sporades | Micro | Greece | 417 | 42 |
| Malta, Gozo, and Comino | | Malta | 316 | 1562 |
| Argosaronicos Islands | | Greece | 261 | 227 |
| Tuscan Islands | | Italy | 261 | 131 |

Source: Observatory on Tourism in the European Islands—OTIE.

Given this evidence, the need to investigate the performances recorded by islands in the tourism sector arises. Comparing the results obtained with those of other islands could be further relevant to defining the best strategies to reach sustainable development through tourism [26,47,48].

## 4. Evolutionary Analysis of the Islands

If we consider hotels and other facilities, the Mediterranean islands counted 24,416 (2019) accommodations and 1,813,269 beds. The distribution of the tourist supply is not uniform in all the islands; one should think that, for example, the Balearic Islands on their own contribute to 25.8% of the total availability in the Mediterranean islands in terms of beds. The Spanish archipelago is the first, counting more beds than Sardinia and Sicily, although characterized by a territorial extension equal to one-fifth of Sicily, which is the largest Mediterranean island.

The highest proportion of tourist accommodation structures is recorded in Sicily (30.6%), followed by Sardinia with 23.4% of buildings being accommodation establishments.

A further comparison can be made by considering the size of the structures. The hotel accommodation class provides the highest number of beds (1,355,348, in 2019) throughout the Mediterranean, albeit with apparent differences from island to island. The largest hotels are in the Balearic Islands, the Maltese Archipelago, the Dodecanese Islands, Sardinia, Crete, Cyprus, and the Ionian Islands. These contexts have hotels that provide an average of no less than 100 beds, according to a range between 264 beds in the Balearic Islands and 103 in the Ionian Islands. Table 2 summarises supply composition in 2007 and 2019.

**Table 2.** Tourist supply in the European Islands.

| Islands | Hotels | | Hotel Beds | | Other Accommodations | | Other Accommodations Beds | |
|---|---|---|---|---|---|---|---|---|
| | **2007** | **2019** | **2007** | **2019** | **2007** | **2019** | **2007** | **2019** |
| Sicily and small islands | 1192 | 1328 | 114,583 | 125,780 | 2562 | 6145 | 66,828 | 85,143 |
| Sardinia and small islands | 846 | 925 | 97,158 | 110,015 | 1875 | 4792 | 92,081 | 107,319 |
| Cyprus | 735 | 814 | 87,804 | 89,200 | 167 | 2 | 4765 | 988 |
| Corse | 367 | 438 | 21,752 | 25,138 | 250 | 451 | 110,161 | 138,892 |
| Crete | 1509 | 1619 | 146,955 | 187,599 | 16 | 15 | 2815 | 760 |
| Balearic Islands | 1393 | 1410 | 326,028 | 371,801 | 1138 | 1362 | 108,229 | 95,925 |
| Northeastern Aegean Islands | 403 | 387 | 20,967 | 23,006 | 1 | 1 | 285 | 46 |
| Evia | 225 | 245 | 15,413 | 16,832 | 8 | 10 | 2180 | 667 |
| Ionian Islands | 897 | 980 | 85,098 | 101,405 | 27 | 24 | 6549 | 1866 |
| Dodekanisa | 972 | 1064 | 120,540 | 167,644 | 3 | 4 | 396 | 152 |
| Cyclades | 942 | 1090 | 42,316 | 56,037 | 31 | 30 | 8888 | 2513 |
| Sporades | 161 | 149 | 10,667 | 10,921 | 0 | 1 | 0 | 72 |
| Malta | 160 | 224 | 39,985 | 46,350 | 6 | 20 | 844 | 1746 |
| Argosaronicos | 160 | 197 | 7051 | 8081 | 0 | 1 | 0 | 60 |
| Tuscan Islands | 210 | 198 | 16,007 | 15,539 | 251 | 490 | 20,869 | 21,772 |

Source: Observatory on Tourism in the European Islands—OTIE.

The other accommodation facilities are smaller than the previous one, except for Cyprus and Corse, equipped with a small number of large structures with an average size of 494 beds and 308 beds in each establishment. This figure is not surprising since the main kind of other facilities in these contexts is camping. The number of establishments and beds is not enough to explain the actual development of tourism in the Mediterranean islands.

In 2019, the tourism flow which affected the Mediterranean islands totaled 43,819,664 arrivals, +53% compared to 2007, and 215,899,617 overnight stays, +34% compared to 2007. Additionally, the demand flow distribution was not equal in all the contexts examined, as highlighted by the fact that 52% of arrivals are to three places (Balearic Islands, Sicily, Crete), and 56% of overnights can be attributed to the Balearic Islands, Crete, and the Dodecanese. In both components of demand, the superiority of the Spanish Archipelago arises. Indeed, it represents almost 30% of arrivals to Mediterranean islands and 32% of the total overnight stays corresponding to more than 68 million nights. It leaves behind all the other areas with a significant gap. Indeed, Crete, the second in overnight stays, has a deficit from the Spanish Archipelago of more than 7,000,000 arrivals, Sicily, which has the second-highest number of admissions, is separated from the first position by more than 50,000,000 nights (Table 3).

**Table 3.** Tourist demand variation in the European Islands, 2007/2019.

| Islands | Arrivals | | | | Overnights | | | |
|---|---|---|---|---|---|---|---|---|
| | **2007** | **2019** | **2019 2007** | **Var.%** | **2007** | **2019** | **2019 2007** | **Var.%** |
| Sicily | 4,614,338 | 5,120,421 | 506,083 | 11% | 14,602,145 | 15,114,931 | 512,786 | 4% |
| Sardinia | 2,280,173 | 3,444,058 | 1,163,885 | 51% | 11,851,213 | 15,145,885 | 3,294,672 | 28% |
| Cyprus | 2,325,608 | 3,242,957 | 917,349 | 39% | 14,377,667 | 17,573,684 | 3,196,017 | 22% |
| Corse | 2,016,110 | 2,901,518 | 885,408 | 44% | 6,240,956 | 10,675,065 | 4,434,109 | 71% |
| Crete | 2,237,139 | 5,048,131 | 2,810,992 | 126% | 15,324,936 | 28,006,885 | 12,681,949 | 83% |
| Balearic Islands | 9,416,695 | 12,425,741 | 3,009,046 | 32% | 62,166,198 | 68,376,034 | 6,209,836 | 10% |
| NE Aegean Islands | 327,188 | 402,581 | 75,393 | 23% | 1,659,124 | 2,007,723 | 348,599 | 21% |
| Evia | 238,463 | 307,871 | 69,408 | 29% | 1,174,998 | 1,209,719 | 34,721 | 3% |
| Ionian Islands | 1,117,009 | 2,315,832 | 1,198,823 | 107% | 7,522,757 | 12,917,772 | 5,395,015 | 72% |
| Dodekanisa | 1,681,136 | 3,887,779 | 2,206,643 | 131% | 13,010,561 | 24,579,700 | 11,569,139 | 89% |
| Cyclades | 459,411 | 1,926,589 | 1,467,178 | 319% | 1,679,526 | 6,206,015 | 4,526,489 | 270% |
| Sporades | 107,015 | 154,617 | 47,602 | 44% | 616,782 | 873,707 | 256,925 | 42% |
| Malta | 1,193,033 | 2,022,912 | 829,879 | 70% | 8,082,229 | 9,911,282 | 1,829,053 | 23% |
| Argosaronicos | 77,238 | 149,692 | 72,454 | 94% | 214,805 | 416,518 | 201,713 | 94% |
| Tuscan Islands | 466,624 | 468,965 | 2341 | 1% | 2,980,209 | 2,884,697 | −95,512 | −3% |

Source: Observatory on Tourism in the European Islands—OTIE.

On average, the length of stay across the area was 4.7 days. Some differences should be noted beyond the individual contexts that can be highlighted according to four categories in which the islands have been divided. The average length of stay is quite similar for small islands, micro islands, and medium islands (5 days) and coherent with the general average shown. A lower length of stay on average lower is recorded in more extensive contexts (3.7 days in the large islands). In that respect, a reflection is needed. Let us suppose the result regarding small islands is due in part to the average 3.2 days of Cyclades and 4 days of Evia, excluding them, the category would have an average of 5.6 days. In that case, the large islands appear unable to restrain their guests for longer than the weekend, especially Sicily with its three days. This shows that broader contexts, which would suggest a more significant presence of tourist attractions or a more significant number of sights, things to do, and places to visit, fail to become holiday destinations probably due to a lack of diversification of the supply.

Even within the same year, the flows were not evenly distributed in all island contexts. In general, tourist movements are more concentrated around May to September. Some isolated cases of seasonality extended from April to October, indicating the presence of a type of resort tourism, which makes the Mediterranean one of the favorite locations for the summer holiday.

Overall, both arrivals and nights have registered positive inflections in the islands of the Mediterranean, in the ten years from 2007 to 2019, with increases of 15,000,000 for appearances and more than 50,000,000 for presences.

Evaluating the overall result in the observed years, the scenarios have recorded an increase of 53% in arrivals and 34% in overnight stays, highlighting a tendency towards more numerous but short travel. The best performances have been recorded by the Greek Islands, Malta, and Sardinia, which show an increase greater than 50% in arrivals, and Greek Islands and Corse with an increase greater than 70% of overnights.

## 5. Data Analysis and Results

The first indicator analyzed is the territorial density index. It allows you to assess how many beds are available per km$^2$. The first interesting result concerns the Maltese Archipelago, which shows the highest concentrations of beds on their territory in 2019, followed by the Italian Tuscan Archipelago. Lower index values are found in two Greek islands (the Northeastern Aegean Islands and Evia) and the two large Italian islands. Another indicator to be considered is the occupancy rate. The numerator indicates the number of visitors' overnight stays. The denominator is the potential number of overnights stays, i.e., the total number of available beds in that year.

This index expresses the efficiency of the management in terms of the ability to maximize the occupancy of the accommodation establishment.

Malta has the best value of this rate in both years observed. The last indicator concerning the supply structure considers the average size of the accommodation establishments in each insular context examined. In general, the size of the accommodation establishments is relatively stable from 2007 to 2019. The most significant structures are in Malta, Corse, the Balearic Islands, and Dodekanisa, with an average size higher than 150 beds. The territorial exploitation index measures the pressure on the environment from tourist and resident populations from the demand side. It relates the impact of tourist arrivals and residents to the territory's total area. Its value can be regarded as an indirect measure of stress that tourists and residents carry on the infrastructures of the region (Table 4).

**Table 4.** Selected indicators examined.

| Index | Statistical Indicator |
|---|---|
| Territorial Density Index—TDI | $\dfrac{\text{Beds}}{\text{Surface}\,(\text{km}^2)}$ |
| Occupancy Rate—OR | $\dfrac{\text{Nights}}{\text{Beds}\times 365}$ |
| Average Size of Establishments—AS | $\dfrac{\text{Total number of beds}}{\text{Total number of establishments}}$ |
| Territorial Exploitation Index—TEI | $\dfrac{\frac{\text{Arrivals+Residents}}{\text{km}^2}}{100}$ |

Comparative analysis of tourist industry/performances in Mediterranean islands requires a selection and evaluation of a set of indicators for all the territorial contexts considered. The table below shows, for each island, the values of the chosen indicators in the two years observed.

Table 5 shows the variation of 2019/2007 for each observed statistical indicator for each island cluster. Corsica, Cyprus, and the Tuscan Islands reduced the territorial exploitation index and, therefore, tourist pressure on the territory. The TEI indicator shows a general increase in territorial pressure in all the islands. The most significant increase is for the Dodecanese, the Cyclades, the Ionian Islands, and Sardinia. In terms of occupancy rate, the Cyclades recorded the best increase in the observed period. The concentration of beds is relatively stable, except that for Dodekanisa which shows a higher density and a greater average size in 2019 than in 2007.

**Table 5.** Tourist indexes in the European islands.

| Islands | Island Group | TDI 2007 | TDI 2019 | OR 2007 | OR 2019 | AS 2007 | AS 2019 | TEI 2007 | TEI 2019 |
|---|---|---|---|---|---|---|---|---|---|
| Sicily | Large | 7.06 | 8.21 | 0.22 | 0.20 | 48.32 | 28.22 | 3.76 | 3.90 |
| Sardinia | | 7.86 | 9.02 | 0.17 | 0.19 | 69.55 | 38.02 | 1.64 | 2.10 |
| Cyprus | | 10.01 | 9.75 | 0.43 | 0.53 | 102.63 | 110.52 | 3.36 | 4.80 |
| Corse | Medium | 15.20 | 18.90 | 0.13 | 0.18 | 213.80 | 184.51 | 2.64 | 3.74 |
| Crete | | 18.13 | 22.80 | 0.28 | 0.41 | 98.21 | 115.27 | 3.44 | 6.88 |
| Balearic Islands | | 87.41 | 94.15 | 0.39 | 0.40 | 171.58 | 168.73 | 21.02 | 27.40 |
| Northeastern Aegean Islands | | 4.99 | 5.41 | 0.21 | 0.24 | 52.60 | 59.41 | 1.24 | 1.48 |
| Evia | | 4.80 | 4.78 | 0.18 | 0.19 | 75.51 | 68.62 | 1.21 | 1.36 |
| Ionian Islands | Small | 37.51 | 42.27 | 0.22 | 0.34 | 99.19 | 102.86 | 5.51 | 10.31 |
| Dodekanisa | | 50.54 | 70.12 | 0.29 | 0.40 | 124.04 | 157.11 | 7.85 | 17.05 |
| Cyclades | | 22.59 | 25.83 | 0.09 | 0.29 | 52.62 | 52.28 | 2.52 | 9.03 |
| Sporades | | 25.58 | 26.36 | 0.16 | 0.22 | 66.25 | 73.29 | 2.94 | 4.12 |
| Malta | Micro | 129.21 | 152.20 | 0.54 | 0.56 | 245.96 | 197.11 | 50.74 | 79.64 |
| Argosaronicos | | 27.02 | 31.19 | 0.08 | 0.14 | 44.07 | 41.12 | 4.98 | 8.01 |
| Tuscan Islands | | 141.29 | 142.95 | 0.22 | 0.21 | 79.99 | 54.23 | 19.08 | 19.27 |

Source: Observatory on Tourism in the European Islands—OTIE.

Since the obtained values are quite different from each other, we proceed with standardizing the data based on the territorial extension of the islands and the maximum value recorded for each index. This normalization leads to a more equal comparison between the different contexts with several structural differences. They are normalized on the maximum value recorded for each island cluster dimension. It does not express the maximum value of the indicator in absolute terms.

The island comparative analysis considers the relationship between tourist pressure on the destination, measured by the territorial exploitation index (TEI), with the three other statistical indicators concerning the structural endowment. This allows analyzing the positioning with respect to the two dimensions simultaneously observed. The graphs show the positioning of the islands according to the relations observed between the two statistical indicators in 2007/2019. The first graph (Figure 1) compares the relationship and evolution between TEI and the occupancy rate (OR). Low TEI characterizes the second quadrant's optimal positioning, where the high OR levels indicate an excellent tourism industry performance and a contained pressure on islands. Considering a dynamic view, the data show a general trend toward the first quadrant: an increase in efficiency, as the rate of beds occupancy increases, with a reduction in sustainability expressed by the rise in the pressure on the island. This is especially true in the case of Sardinia and the Dodecanese, which pass from the second to the first quadrant during the observed period. Although it remains in the first quadrant, Cyprus significantly reduced its tourist pressure while maintaining production efficiency. In 2019, the Cyclades Islands improved both parameters and moved from the third to the second quadrant.

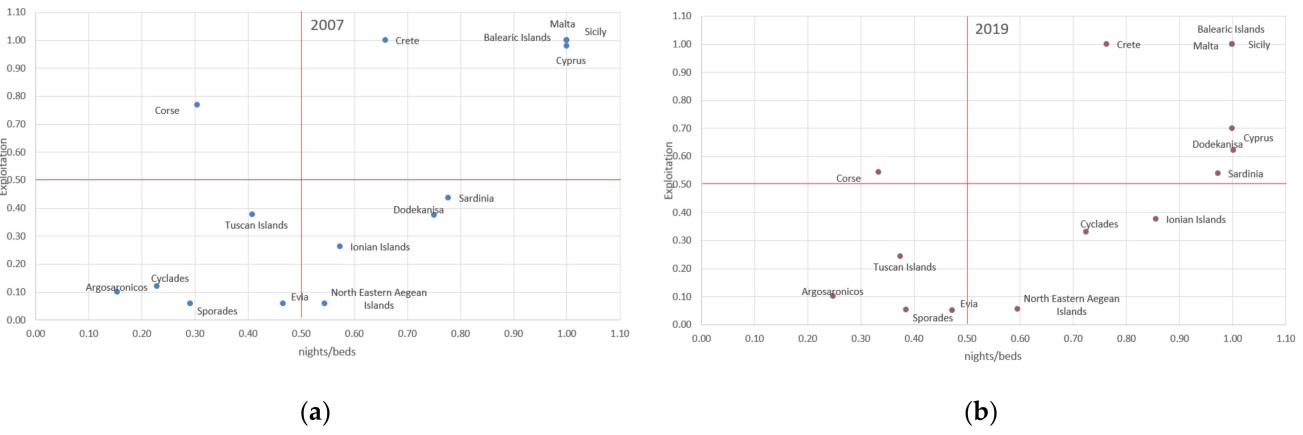

(**a**)　　　　　　　　　　　　　　　　　　　　　　　　　　　　　(**b**)

**Figure 1.** Territorial exploitation index and occupancy rate in 2007 (**a**) and in 2019 (**b**). Source: Observatory on Tourism in the European Islands—OTIE.

Precisely, small and micro islands lie in the second and third quadrants both in 2007 and 2019, at least moving within them. For example, the Tuscan Islands and the Ionian Archipelago improve their results by reducing the TEI value. Among these islands, the only exception is the Dodekanisa, which moves from the second quadrant (the best position) to the first one, getting worse in sustainability. The Balearic Archipelago and Malta maintain Sicily's position, the largest Mediterranean island.

Large and medium islands are between the first and the last quadrant, highlighting less attention to socio-environmental issues. Cyprus and Corse reduced the TEI from 2007 to 2019, while Sardinia moved from the best quadrant to the first one, improving efficiency at the expense of sustainability.

The second analysis (Figure 2) concerns the relationship between the TEI and the structural characteristics of the tourism supply, described by the average size indicator (AS) of the accommodation facilities. In this case, the desirable positions are the second and third quadrants, connected to a low socio-environmental impact. Dodecanese and Sardinia worsen the performance related to environmental pressure and move from the second to

the first quadrant. In parallel, in 2019, Corse Island showed more significant attention to sustainability positioning almost at the border between the first and the second quadrant. Moreover, Cyprus improved its performance by reducing the socio-environmental impact, as revealed by moving from the top side of the last towards the lower part of the first quadrant.

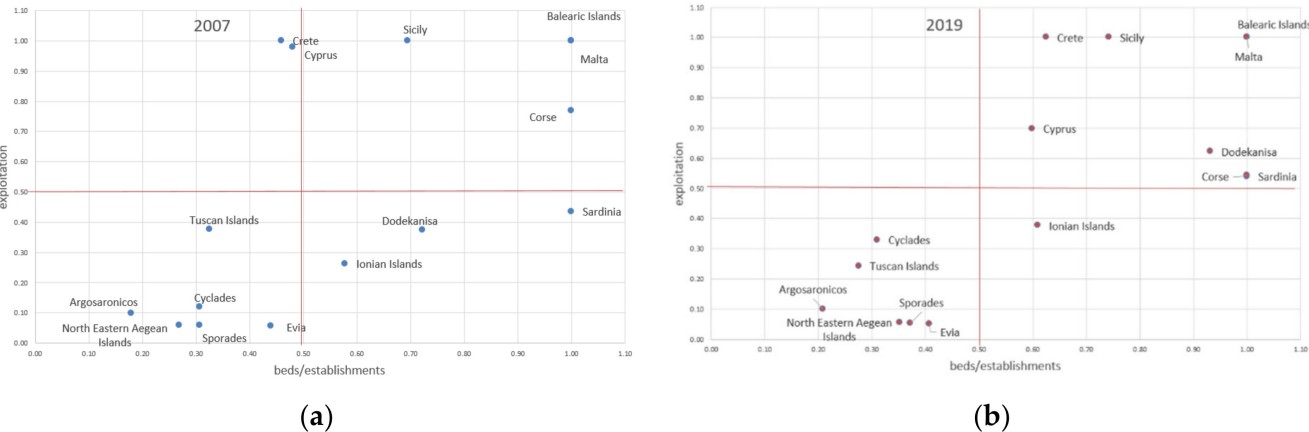

**(a)**                                                      **(b)**

**Figure 2.** Territorial exploitation index and the average size of accommodation establishments in 2007 (**a**) and in 2019 (**b**). Source: Observatory on Tourism in the European Islands—OTIE.

In general, as in the previous example, the large and medium islands have the worst positioning in the first quadrant, with a high average size of the accommodation facilities and an equally high value of the TEI. Additionally, Sardinia, which in 2007 was in the second quadrant, in 2019 moved into the first one. Small and micro insular contexts lie in the second and third quadrants both in 2007 and 2019, except for the Spanish Archipelago and Malta, which have the exact same position as the big islands.

Figure 3 compares TEI with the territorial density index (TDI), which is concerned with the territorial concentration of tourist supply in terms of beds. The best positioning is the third quadrant, characterized by low beds on the territory and low-pressure levels.

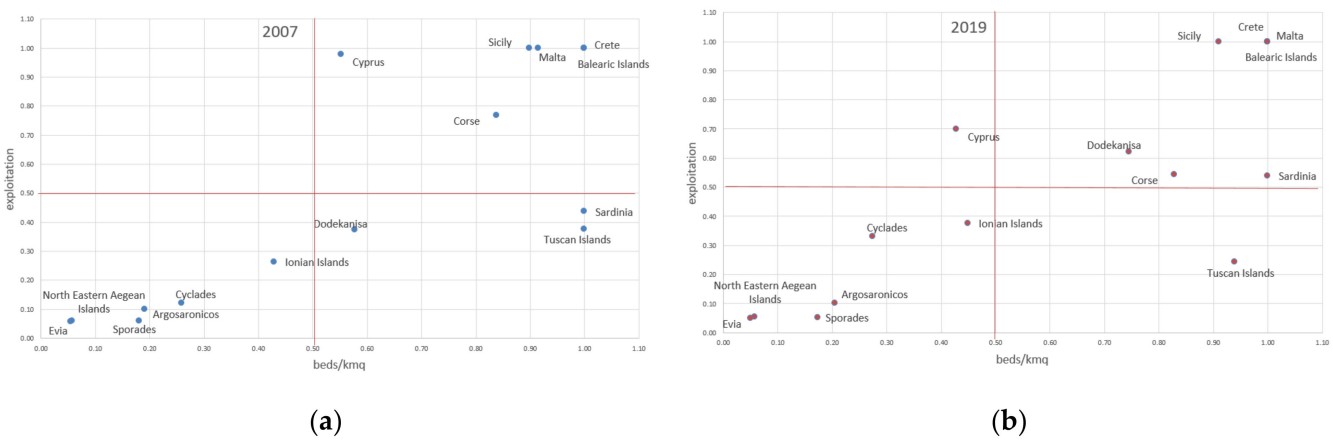

**(a)**                                                      **(b)**

**Figure 3.** Territorial exploitation index and the average size of accommodation establishments in 2007 (**a**) and in 2019 (**b**). Source: Observatory on Tourism in the European Islands—OTIE.

The small and micro islands have the lowest values of the TEI and are in the second and third quadrants, except for Malta and Balearic, which follow the large islands. Dodekanisa, among the small contexts, and Sardinia, among the large ones, move from the second to the first quadrant and both worsen their position in terms of density of beds and TEI. Cyprus improves its performance moving from the top side of the first quadrant to the lower part of the last one, with a lower TEI and density of beds for square kilometers. Corse stays

within the first quadrant reducing socio-environmental impact, moving towards the lowest part of the area. Six island contexts take up the win–win position in the two observed years (third quadrant). These are small and micro contexts, highlighting once again greater environmental attention than larger ones.

## 6. Discussions: Policy Implication for Mediterranean Islands

Fifteen insular contexts belonging to six different countries, Cyprus, Greece, Italy, Malta, France, and Spain, were compared to highlight general findings and specific features.

Insular contexts are different in geo-demographic and institutional dimensions and in terms of tourism development.

The various combinations of territorial extension, population, and tourism industry characteristics lead to different socio-environmental impacts and levels of efficiency in managing the tourism industry in two different periods of time.

The distribution of the tourist supply is not uniform across all the islands. The Spanish Archipelago is the first in terms of beds, counting more beds than Sardinia and Sicily, although characterized by a territorial extension equal to one-fifth of Sicily, which is the largest Mediterranean island. Here we find the highest portion of tourist accommodation structures (30.6%), followed by Sardinia with 23.4%.

Considering the size of the structures, the highest number of beds is in hotel accommodations (1,355,348, in 2019). With more than 100 beds, the largest hotels are in the Balearic Islands, the Maltese Archipelago, the Dodecanese Islands, Sardinia, Crete, Cyprus, and the Ionian Islands.

The other accommodation facilities are smaller than the previous one, except for Cyprus and Corse, equipped with a small number of large structures with an average size of 494 beds and 308 beds per establishment. This figure is not surprising given that the main kind of other facilities in these contexts is camping. In 2019, both arrivals and overnights increased in the islands of the Mediterranean Basin (+53% and +34%, respectively). Remarkably, 52% of arrivals are due to the Balearic Islands, Sicily, and Crete, and 56% of overnights can be attributed to the Balearic Islands, Crete, and the Dodecanese. The Spanish Archipelago, in itself, represents almost 30% of arrivals to Mediterranean islands and 32% of the total overnight stays corresponding to more than 68 million nights. Considering the variation in the observed period, the best performances have been recorded by the Greek Islands, Malta, and Sardinia, which show an increase greater than 50% in arrivals, and the Greek Islands and Corse with an increase greater than 70% for overnights.

Malta shows the highest TEI and TDI values in sustainability and socio-environmental impact.

By focusing on the deviations recorded by each index during the period 2007–2019, the best and worst cases can be highlighted. Corsica, Cyprus, and the Tuscan Islands show to have reduced the territorial exploitation index and, therefore, tourist pressure on the territory. The islands that experienced the most significant increase in this indicator are the Dodecanese, the Cyclades, the Ionian Islands, and Sardinia. In terms of occupancy rate, the Cyclades recorded the best increase in the observed period (+0.5). The concentration of beds is relatively stable, except that Dodekanisa, showing a higher density (+0.17) and a greater average size (+0.21) in 2019 than in 2007.

Comparing the TEI index with the other three indicators selected, greater attention to sustainable aspects in the small contexts can be observed. Large islands always appear in the quadrant corresponding to the higher socio-environmental pressure.

In general, the Cyclades, the Ionian Islands, and the Northeastern Aegean Islands are always in the win–win quadrant. On the other hand, large and medium insular contexts always show low sustainability positions. Balearic and Malta, among the small and micro contexts, show the same positioning. Sardinia began with a sustainable approach in 2007, moving towards the first quadrant in 2019, getting worse in terms of socio-environmental impact.

### 7. Conclusions: Islands' Tourism Policy Implications

Islands are considered fragile territories due to the limited physical and economic resources and an unstable environmental balance. Sustainability aspects are always regarded as central for those territories, and at the same time, the need to support local economies through tourism is considered essential. The paper compared Mediterranean islands' performances by using statistical indicators considering island clusters. The analysis shows that islands are characterized by a model of tourist development that has encouraged the construction of large hotels with a high average number of beds per establishment, thus creating sizable and prominent tourist destinations.

The need to increase the number of tourism establishments, number of beds, and the need to rise in efficiency measured by beds occupancy resulted in a rise in island pressure between 2007/2019. Analysis results are more evident for large and medium Mediterranean islands and in the case of large archipelagos. Due to this comprehensive tourism policy, the pressure on the islands is increasingly attracting more visitors to islands with an increase in tourists and overnights. Conversely, small and micro islands kept a contained pressure in 2007/2019 by choosing a small establishment dimension.

The analysis could further consider other external factors that influenced the increase of tourist supply: territorial dimensions, ability to attract investment, size of flows, and different time stages of these destinations' life cycles.

From the results obtained, island dimensions show a natural limitation in tourism investments. Large and medium islands and archipelagos offer a development model based on the tourism industry model, increasing the industry, following the increase in tourism demand before the COVID-19 pandemic. The rise in island pressure was not considered a limitation, and the expansion of the market supported the economic growth in the industry and local economy. Small and micro islands followed a more balanced model, by following the demand increase which adopted policies to keep a moderate level of pressure and islands sustainability.

The analysis represents a starting point for further studies and insights. Mediterranean islands need to address strategic development policy to ensure economic efficiency and at the same time respect the local environment and culture. In this context, new technologies, as well as European strategies, could support the management to take action on specific issues, like urban and environmental planning, mobility, smart cities, waste, and water management, energy consumption, promotion of local culture, and tourist flow management.

Furthermore, advances in ICT help improve destination management and promotion at the same time raise visitors' awareness towards tourism that respects local people and resources [49].

**Author Contributions:** The authors equally contribute to each section of the paper. Particularly, G.R.: Conceptualization, Methodology, Validation, investigation, resources, Writing—Original Draft, Writing—Review & Editing, Supervision; P.C.: Conceptualization, Methodology, Formal analysis, resources, data curation, Writing—Original Draft, Writing—Review & Editing, visualization. All authors have read and agreed to the published version of the manuscript.

**Funding:** This research received no external funding.

**Institutional Review Board Statement:** Not applicable.

**Informed Consent Statement:** Not applicable.

**Data Availability Statement:** Data supporting reported results can be found at Eurostat Database https://ec.europa.eu/eurostat/data/database/ (accessed on 10 January 2022).

**Conflicts of Interest:** The authors declare no conflict of interest.

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
