# Peer review of "Tourism Dynamics and Sustainability: A Comparative Analysis between Mediterranean Islands—Evidence for Post-COVID-19 Strategies"

_sustainability, doi:10.3390/su14074183_

Round 1

Reviewer 1 Report

The analytical part of the article is fine, but it severely lacks contextualization and clarification of the used key concepts and terms. 
For Example: What is the definition for tourists that you use? How do you ensure that (even if small in number) business guests are not counted as well? There are well established scientific concepts for the research question you are addressing. Carrying Capacity for Example. Why do you not use it? It should at least be explained why you are not using it. Sustainability and also Sustainable Development has many dimensions. It should be elaborated better which dimensions you are adressing? How do you link it to SDGs for Example? You mention sustainable tourism several time. How do you define it? There have been several European initiatives and Guidelines for sustainable tourism. How does your research relate to them? It seems that one question you are examining is also related to the Quality of Life of the islands inhabitants. There are several schemes to evaluate Quality of Life and it would enrich your paper to relate your number the these and draw conclusions that are better rooted in the perception of the inhabitants. It would also be interesting to learn more about you conclusions on what could actually be done in addition to ideas for further research. Alltogether the topic is very relevant, but without a better contextualization and more clarity about the incorporated context the current state of the paper is underdeveloped and it is not ready for publication. 

Reviewer 2 Report

Try to better define the aspects of critical tourism and overtourism, making greater use of the extensive existing literature 

Reviewer 3 Report

Dear Authors,

Very good, well-written and professional paper.  

Good luck!

Round 2

Reviewer 1 Report

Now the article is ready for publication